# “The Ugliness of It Seeps into Me”: Experiences of Vicarious Trauma among Female Psychologists Treating Survivors of Sexual Assault

**DOI:** 10.3390/ijerph19073925

**Published:** 2022-03-25

**Authors:** Anita Padmanabhanunni, Nondumiso Gqomfa

**Affiliations:** Department of Psychology, University of the Western Cape, Cape Town 7535, South Africa; nondumiso7@gmail.com

**Keywords:** psychologists, phenomenology, sexual assault, South Africa, survivor guilt, treatment, vicarious trauma

## Abstract

The current study explores the lived experiences of female psychologists who provide psychological treatment to women survivors of sexual assault. These practitioners are a population of special interest due to the frequency of their exposure to narratives and graphic images of sexual trauma. Semi-structured interviews were conducted with 15 female South African psychologists. The data, which were analyzed using an interpretative phenomenological approach, revealed experiences characteristic of vicarious traumatization. Participants reported having an enhanced sense of personal vulnerability to sexual assault and heightened awareness of the betrayals of trust that women experience from male figures, which has led to increased mistrust of men and hypervigilance regarding the safety of their daughters. Internalization of feelings of helplessness experienced by the victim evoked self-blame for practitioners and appraisals of being complicit in abuse. Practitioners also experienced survivor guilt about being spared from harm. Symptoms of intrusive re-experiencing of client trauma and cognitive and behavioral disengagement were evident. These findings have important implications for clinical practice and underscore the necessity for clinicians to cultivate an awareness of the impact of treating sexual trauma and working in ways that are self-protective.

## 1. Introduction

Sexual assault perpetrated against girls and women is a pervasive problem globally and has been associated with a range of adverse mental and physical health outcomes for survivors. This includes post-traumatic stress disorder (PTSD), depression, substance use, anxiety, suicidality, and negative reproductive health outcomes [1]. Women who have experienced sexual assault report significant feelings of hopelessness, loneliness, overwhelming anxiety, and difficulty functioning in their daily lives. Sexual assault is more strongly associated with suicidal behavior than other types of traumatic events [2].

The current study was undertaken in South Africa, which has the highest global rate of sexual offences against women and girls [3]. Estimates of the prevalence of sexual assault in South Africa have varied due to survivors’ hesitance to disclose, differences in the perpetration of rape across various locations in the country, the nature of the scale used to measure exposure to sexual assault, and characteristics of the population sampled [4]. A nationally representative survey of school children [5] found that 14.6% of girls aged 15–17 reported that they had experienced sexual assault. Cross-sectional studies of university students (e.g., [6]) have reported prevalence rates of sexual violence against women ranging from 24 to 32.2%. In a household survey conducted in the North West province of South Africa, [4] reported a lifetime sexual violence prevalence rate of 24.9%. Nearly one-third of participants stated that they had not disclosed their sexual assault because they were financially dependent on the perpetrator. The prevalence of sexual assault can also be deduced from male perpetrator prevalence studies, which indicate that 28–35% of men report having committed a sexual assault [7].

Studies conducted in South Africa have confirmed that sexual victimization is strongly associated with PTSD, depression, and suicidal behavior [7,8,9]. Many victims seek formal mental health care to address the emotional distress of sexual assault and its adverse impact on relationships and academic and occupational functioning [1]. There is evidence to suggest that sexual assault victims prefer to seek help from a female mental health care provider, possibly due to the gendered nature of sexual trauma and the belief that women are more likely than men to be empathic, non-judgmental, and supportive [10]. The bulk of the literature related to sexual assault has focused on the experiences of the primary victim. To date, little research has explored the psychological impact of working with sexual assault victims on female mental health care providers [11].

Long-term and repeated exposure to patients’ traumatic lived experiences can cause disturbances in a clinician’s cognitive schemas, including their beliefs that the world is safe and predictable, that people are generally benevolent, and that the self is worthy and protected [12]. These disturbances in cognitive schemas are associated with changes in self-identity, worldview, and spirituality, and they are characteristic of vicarious traumatization. In the seminal paper of [13], the authors describe vicarious trauma as the profound and lasting emotional and psychological consequences of repeated indirect exposure to the traumatic experiences of others. Vicarious trauma arises from empathic engagement with traumatic material, including graphic narrative descriptions of violent experiences and exposure to the reality of people’s cruelty toward each other [14]. Vicarious trauma reactions include symptoms characteristic of PTSD, such as intrusive thoughts and images of the victim’s trauma, avoidance and emotional numbing, and symptoms of hyperarousal [11].

A significant body of research has documented vicarious trauma among different categories of medical care workers (e.g., nurses: ref. [15]; forensic medical examiners: ref. [16]) and mental health care providers (e.g., psychotherapists: ref. [17]; social workers: ref. [18]; psychiatric hospital workers: ref. [19]). Additionally, an increasing number of studies have investigated the impact of working with sexual assault victims for law enforcement personnel [20], rape medical advocates [21], sexual assault nurse examiners [22], and social workers [23]. Female mental health care providers who work predominantly with sexual assault cases have been found to experience more disrupted beliefs about themselves and others than their peers. For example, they may experience altered beliefs regarding the goodness of other people, an increased sense of personal vulnerability, and concerns about personal safety and the safety of significant others, as well as difficulties with sexual intimacy [24,25]. In addition, female mental health care providers who work with survivors of sexual assault are more likely than their peers to experience symptoms characteristic of PTSD [12].

Psychologists who treat survivors of sexual assault are more susceptible to vicarious trauma compared to mental health providers who treat other types of traumas [22]. The potential effects of working with trauma survivors in the context of psychotherapy are distinct from those associated with other professions [11]. The clinician acts as an ally and witness to guide victims to revisit and construct a narrative around their trauma. Through this process, the clinician is exposed to the emotionally charged, graphic images of violence and suffering that are characteristic of sexual trauma [12]. Further, psychological interventions for treating intrusive memories associated with PTSD involve imaginal reliving, where the clinician guides the survivor to relive the event while simultaneously describing what is occurring and narrating their thoughts and feelings at the time [26]. The aim of this intervention is to facilitate cognitive processing and identify and address any problematic appraisals (e.g., self-blame) associated with the trauma. Imaginal reliving exposes the clinician to the subjective lived experience of sexual assault in a way that is distinct from the experiences of other professionals who work with rape victims.

The current interpretative phenomenological study aimed to explore the lived experiences of female psychologists in South Africa who provide trauma therapy to women survivors of sexual assault and to understand the ways in which they are impacted by their work. The use of an interpretive phenomenological approach is an advantage of this study as it promotes deeper levels of reflection for the researcher, enhances psychological depth in the interpretation of the subjective experiences of the participants, and facilitates the deepening of our contextual awareness of the phenomenon under study [27].

## 2. Materials and Methods

### 2.1. Research Design

The current study used interpretative phenomenological analysis (IPA [27]) to explore and describe the subjective lived experiences of clinicians. IPA focuses on lived experience through case-by-case analysis and empathic engagement with the participants’ narrative accounts. This approach was selected for the study because it facilitates in-depth inquiry and detailed examination of real-life experiences.

### 2.2. Participants

In accordance with the sampling principles of IPA, a homogenous sample of participants was selected. Participants were 15 female clinical and counselling psychologists who predominantly provided therapy to survivors of sexual assault. The mean age of participants was 38 years, and the average number of years in the profession was 6.5 years. Twelve participating clinicians reported a history of personal direct exposure to traumatic events, and two reported a prior history of sexual victimization. Ten participants were Black South Africans and five were Caucasian.

### 2.3. Data Collection

Participants were recruited by advertising the study on professional networks for psychologists. Respondents completed informed consent forms. Semi-structured interviews of 60–90 min in duration were conducted and audio recorded by the second author who then transcribed each interview verbatim. The semi-structured interview protocol included open-ended questions that were informed by the relevant literature on vicarious trauma (e.g., [13,25]). The questions centered on clinicians’ experiences of working with sexual trauma patients and the psychological, interpersonal, professional, and emotional impact of this work.

### 2.4. Data Analysis

The procedural guidelines for IPA established by [27] were used to analyze the data. Individual transcripts were analyzed line by line and coded. Thereafter, a second reading of the transcripts occurred and focused on obtaining an in-depth understanding of the material and identifying and mapping emergent themes. The second author, a Master’s student, undertook the initial coding and identification of themes. Thereafter, the codes and themes were checked against randomly selected transcripts and verified by the first author who was supervising the project. The second author generated the initial superordinate and constituent themes for each participant. These were verified and refined by the first author. Thereafter, patterns and variations in experiences across transcripts were identified by the second author and confirmed by the first author.

### 2.5. Ethics

Institutional review board approval was obtained from the Humanities and Social Sciences Research Committee of the University of the Western Cape (Ethics reference number: HS15/4/50). To minimize the risk of harm, the interview questions were provided to participants a week prior to the interviews. No direct questions about client sexual assault incidents were posed, and participants were provided with counselling resources.

## 3. Results

The analysis resulted in five superordinate themes, each of which is discussed separately and summarized in Table 1.

### 3.1. Shattered Assumptions

All participants in the study reported experiences that reflected disruptions in their pre-existing worldview or core cognitive schemas about themselves, the world, and other people. Working with sexual trauma patients led to an increased sense of personal vulnerability to sexual assault and a mistrust of men, which impacted clinicians’ personal relationships and approaches to parenting.

#### 3.1.1. Increased Sense of Personal Vulnerability to Sexual Assault

Participants’ sense of enhanced vulnerability to sexual assault was evident in their appraisals that harm was inevitable and that they “could be raped at any point”:

“I haven’t been through sexual trauma in my life but now it’s top of my list of fears. It is something that I think about a lot…[It] makes me feel vulnerable, it makes me feel like I am not out of the loop, I am never going to be out of the loop, I could become a statistic any point in time.”“Women are vulnerable, and women get raped. This is our reality, so I expect to get raped.”

#### 3.1.2. Mistrust of Men

Participants reported experiencing increased fear and distrust of men due to their work with victims of sexual assault:
“I’ve become more fearful of men, feeling as if I don’t know who to trust anymore and don’t know what I am really dealing with when I do encounter a male. How do I know? I do not know for sure.”“It made me realize that we don’t live in a safe society, and even people so close to me like a brother, father, and uncle can inflict harm.”

For some participants, increased mistrust of men filtered into their relationships with male partners:
“I don’t hear about good men in this work. I remember I once said to my husband: ‘I’m sure I won’t be surprised at you when you have an affair. I have not yet heard of good men… I will be surprised if you don’t cheat or hurt [me]’…which is unfair to him actually. He has never shown anything that would suggest that—but it’s just that [because of] this work, I forget [that] there are good men out there…”

It also impacted their parenting practices with their daughters, resulting in increased vigilance, including not allowing their daughters to be alone in the care of a man and restricting their activities to avoid potential harm:
“I expect too often that people will do bad things, and this is illustrated in how I parent my daughters. I always tell [my husband] it does not matter if it is her uncle or grandfather—she will not be allowed in the care of just a male because I have seen so many clients that have been raped by their uncle, father, or grandfather.”“My husband and I have decided that we won’t allow sleepovers. I know how quickly these things happen, how the very nice uncle or father or brother or whoever…when the lights go off, all sorts of things happen.”

#### 3.1.3. Recognizing Malevolence in the World

Participants shared that listening to narratives of sexual abuse challenged their assumptions regarding the benevolence of others and evoked feelings of shock and horror that people were capable of perpetrating such harm. As one participant described it, “The level of description of abuse is horrific, I have actually felt repulsed and physically sick.”
“I am shocked and horrified. I feel absolute horror. I mean, I am aware of these things but when [I] hear them firsthand, it is very different…a good word to describe it is absolute horror at how human beings can hurt other human beings in a way I can never get used to.”

Several participants reported experiencing a loss of faith and hope in the goodness of others and feeling “stuck in a place of despair”: “This…it feels like the ugliness of it seeps into me, and sometimes these cases keep me stuck in a place of despair.”

### 3.2. Self-Blame and Survivor Guilt

A salient theme identified in the study was clinicians’ experience of self-blame in relation to the trauma experienced by survivors they treated. This experience included appraisals of “being complicit” in the victimization because they had done nothing to change what had occurred. Through close questioning, it became evident that these appraisals were evoked in the context of exposure-based interventions:
I always feel complicit. I am watching these scenes unfold and just standing there and doing nothing. I was trying my best, but it amounted to nothing. Nothing was changing.

For other participants, guilt was related to appraisals that their patients “lived such hard lives” and suffered rape whereas, in comparison, their lives had been largely protected.
“It brings up guilt and ‘why does this person have to live such a hard life?’ So, there are occasions where at the end of the day I will just walk into my house and start crying.”

### 3.3. Intrusive Re-Experiencing 

All participants reported experiencing intrusions of their patients’ trauma, which were typically triggered by environmental stimuli that bore some resemblance to the traumatic event. One clinician in the study described struggling to bathe her daughters because their naked bodies reminded her of a patient who was sexually abused as a child:
“I could not stop thinking about the image of this woman [as a] five-year-old girl taken into the spare room by her brother and being sodomized. I struggled to bathe my girls because their naked bodies reminded me of this and what might… I struggled with these images that wouldn’t stop.”

Sexual intimacy with partners was another salient trigger for participants. One clinician disclosed that she had developed vaginismus, which she ascribed to the psychological impact of her work on her body: “I had developed vaginismus. Intercourse was very painful, and that was also because I would think about the [patient’s] abuse just before we have sexual relations, or I would get a flashback and that would be extremely difficult for me.”

For other participants, intrusions took the form of nightmares: “My sleep gets affected. I’ll have graphic nightmares. They’re violent, and often something’s happening to me. I’m being raped or abused.”

### 3.4. Emotional and Behavioral Disengagement

Clinicians described emotional and behavioral disengagement due to the impact of their trauma work, including behaviors oriented away from the perceived stressor or their reactions to it (e.g., emotional disengagement) and aimed at reducing distress:
“The most adverse effect is I distance myself emotionally. I am not emotionally available to friends and family as I used to be, I tend to want to avoid them… It is because of my experience and because of the patients I see and the things I hear.”

Avoidance also occurred in the context of their work, including avoidance of wards where they could potentially encounter cases of sexual trauma, ambivalence about working with sexual trauma cases, and reluctance to facilitate a full narrative of the patient’s trauma and therefore prematurely ending assessment interviews:
“I found that I was actually avoiding certain wards in the hospital. I would turn on my heels and not go into particular wards. Some days when I drive into work, I feel a deep sense of dread, there’s part of me that wants to run for the hills. I started noticing that I was cutting [patient] interviews short.”

One clinician described strong physical reactions and discomfort when anticipating working with sexual trauma patients: “I would [get a] physical sensation in my body, a discomfort in my stomach, heart palpitations thinking that I do not want to work with this patient.”

### 3.5. Coping with Vicarious Exposure to Sexual Trauma

Clinicians in the study used both problem and emotion-focused coping strategies to manage the impact of indirect exposure to sexual trauma on their lives. These strategies included modulating their exposure to trauma work, identifying and prioritizing activities outside of work that were fulfilling, attending peer supervision sessions, and positively re-appraising the nature of their work.

#### 3.5.1. Modulating Exposure to Trauma Work

Several clinicians reported the importance of modulating their exposure to cases of sexual trauma by being cognizant of their emotional capacities, setting a limit on the number of sexual trauma cases they saw per day, and spacing sessions each week:
“I am very responsible about taking breaks and not booking patients back-to-back. If I work long hours one day, then the next day I may start a little bit later. I [am] scrupulous around my personal well-being and not burning myself out. I don’t want to feel that I’m giving the patient that I see first a different session to the one that I see last.”

Modulating exposure to trauma also entailed being mindful in the context of therapeutic work and maintaining a differentiation between self and other:
“In the beginning, I [used to] feel completely washed-out at the end of the session. I have realized now that I need to hold back. I need to self-preserve, so even in the session, I remind myself, ‘I’m not them, I’m not going through this. I’m just walking beside them on this part of their journey’.”

#### 3.5.2. Accessing Instrumental Support

Collegial support through individual and peer supervision, as well as personal psychotherapy, was identified as a central strategy to facilitate emotional coping. Supervision provided clinicians with a space to reflect on cases and process the traumatic narratives presented by their patients:
“This work can’t be done without a supervisor because the stories that you hear, you share them for the first time in supervision while it’s raw, and then you can reflect [on] ‘what does it mean to me’ and process it.”

#### 3.5.3. Positive Re-Appraisal

One of the ways clinicians coped with repeated exposure to cases of sexual trauma was by re-appraising trauma work as a “sharing of the load” with the patient, appreciating that surviving trauma requires resilience and strength and viewing their work as healing. These sentiments are captured in the accounts below.
“People come with their trauma, but they bring something else that makes the trauma bearable for the listener. They bear this burden; you share in that load. It’s not just left with you. It’s the sharing of this load… they give you some of how they managed to hold the [trauma], they give you some of that.”“What helps me cope is just to say that ‘what I am doing, is healing’. It is not a mere opening of wounds that are going to be left to fester, but it’s a healing conversation, so my role is to help facilitate healing.”

## 4. Discussion

The aim of this study was to explore the lived experiences of female psychologists who work predominantly with women sexual assault survivors. These mental health care practitioners represent a population of special interest due to the frequency of their exposure to narratives and graphic images of sexual victimization. This exposure is, in part, due to the nature of their work, which entails the use of interventions that involve guiding clients to retell their traumatic experiences in significant detail [28]. These interventions also involve imaginal or in vivo exposure to aspects of the traumatic event, which is intended to help clients address avoidance, process intrusive memories, and target maladaptive appraisals of the trauma and its sequalae [26]. As a result, clinician indirect exposure to traumatic sexual experiences accumulates over time and is likely to have distinctive implications for the clinician’s mental health.

Our findings confirm that working therapeutically with survivors of sexual assault challenges the assumptive worldview of female clinicians in ways emblematic of vicarious traumatization. First, clinicians reported becoming highly sensitized to the reality of sexual violence and the extent to which women are vulnerable to victimization. This sensitization translated into increased cynicism, a heightened sense of personal vulnerability, and loss of a sense of safety in the world, all of which are evident in participants’ appraisals that rape was inevitable and that they “expected to be raped” at any point. According to cognitive theories of stress and coping [20,28], core schemas develop during childhood through interactions with significant others. The most central of these global patterns of belief are the notions that the world is predictable and meaningful, other people are benevolent, and the self is worthwhile and protected. The primary function of these cognitive assumptions is to promote adaptive coping by providing the individual with meaning, self-esteem, and the illusion of invulnerability. The experience of trauma severely disrupts and challenges these global belief systems, and this incongruity creates internal dissonance and psychological distress [28].

Processing traumatic material requires re-appraisal and revision of pre-existing cognitive schemas that are no longer tenable [28,29]. This process occurs either through assimilation, in which the experience of trauma is interpreted in a manner that is consistent with pre-existing cognitive schemes, or accommodation, in which schemas are transformed to incorporate the trauma. Appraisals of traumatic events as being inevitable are reflective of cognitive over-accommodation, which occurs when trauma-related information is overgeneralized. Such over-accommodation can lead to heightened fear, hypervigilance, and extreme measures taken to avoid perceived exposure to harm [28,30]. The experiences of the clinicians in this study may be reflective of over-accommodation; however, these responses may also be adaptive in a social context in which the likelihood of sexual victimization is high. These types of appraisals may reflect participants’ recognition of a reality in which sexual assault is in fact normative and hypervigilance to potential threat is an effective means of self-protection.

Second, this study found that working with sexual trauma cases acutely sensitized clinicians to the betrayals of trust that women experience in their lives, particularly from male figures. This sensitization led to increased mistrust of men, which impacted clinicians’ intimate relationships with male partners and heightened their vigilance about the safety of their daughters. This hypervigilance manifested in avoiding leaving their daughters solely in the care of male figures, including male family members, and restricting their activities to avoid potential exposure to sexual victimization. Similar alterations in cognitive appraisals of trust have been reported in the literature among sexual assault counsellors [31], nurse examiners [22], and mental health care providers who work with sexual offenders [32].

Third, a distinctive finding of this study was the clinicians’ experience of self-blame related to the trauma of their patients. Self-blame is common to the experience of trauma in general; however, disproportionate levels of self-condemnation have been documented among victims of rape [33]. According to [28], self-blame reflects an attempt to reconcile the experience of trauma with one’s pre-existing assumptions about a meaningful world in which there is a comprehensible person–outcome contingency (i.e., the belief that events are not random and there is a relationship between a person and what happens to them). By blaming oneself, the victim can retain illusory assumptions about mastery and control (i.e., if they had done something differently, their victimization could have been prevented). Self-blame is also influenced by societal victim-blaming attitudes and rape myths that hold the victim accountable for their trauma. Acceptance of these attitudes is associated with heightened self-blame and strong feelings of guilt and shame [33].

For the clinicians in the study, self-blame manifested in appraisals of “being complicit” in their clients’ victimization because they were not able to prevent it. Notably, these cognitions were evoked in the context of interventions that involved retelling of the trauma narrative and therefore may be related to transference dynamics (i.e., the displacement or unconscious projection onto the clinician of the client’s feelings) and the internalization of the sense of helplessness and powerlessness experienced by victims [12]. Clinicians also reported experiencing guilt related to social comparisons with the experiences of their clients. These types of appraisals are reminiscent of the construct of survivor guilt or inequity guilt. They include feelings of guilt about being spared from the harm others have experienced, as well as guilt related to appraisals of having any form of advantage (e.g., better health, good relationships) compared to others [34].

Fourth, a common theme reported by clinicians was the intrusive re-experiencing of their patients’ trauma in the form of unwanted thoughts, images, and nightmares. These intrusions were often triggered by sexual intimacy with their partners or situations that bore some similarity to the client’s experiences (e.g., one clinician reported that bathing her young daughters triggered intrusive images about a client who was victimized as a child). Intrusive re-experiencing is a core symptom of vicarious trauma that has been well documented in the literature (e.g., [12]). To cope with their distress, clinicians reported engaging in defensive practices such as avoidance of sexual intimacy, emotional disengagement, and avoidance of working with cases of sexual trauma.

In addition to identifying and detailing the potentially adverse effects of working therapeutically with survivors of sexual trauma, it is equally important to understand the ways in which clinicians sustain themselves. Clinicians in the current study reported re-framing their work as a means to promote healing and as bearing witness to the capacity of others to overcome pain and adversity. These types of appraisals are reminiscent of the construct of vicarious post-traumatic growth, which refers to enduring positive changes in the practitioner’s sense of self and life philosophy [35]. In addition, clinicians reported using a range of emotional and problem-focused coping strategies to manage the impact of their work. These strategies were aimed at self-protection and included modulating their exposure to trauma work by setting limits on their caseloads, taking frequent breaks, having a diversity of cases to prevent saturation, and distributing client loads in a way that was manageable. Peer supervision was consistently identified as central in promoting their capacity to work in this field.

Trauma-informed supervision has been identified as a central protective factor for those involved in trauma work [11,18]. In South Africa, trauma-informed supervision must be grounded in an awareness of the distinctive phenomenology of vicarious trauma in this setting. This includes attentiveness to specific types of vicarious trauma-related appraisals (e.g., clinician self-blame, sense of complicity, and inequity guilt), coping responses (e.g., adaptiveness of hypervigilance in a context in which sexual violence is highly prevalent), and encouragement of clinicians to adopt a self-care plan and facilitate a safe and respectful collaborative relationship. The regenerative supervision model is a culturally responsive framework for supervision that may be particularly beneficial for trauma practitioners due to its focus on enhancing clinician awareness, empowerment, meaning making, and authentic expression of intersession dynamics [36].

This study expands on existing research (e.g., [12]) that has used a phenomenological approach to explore the experience of trauma work among mental health care practitioners. For example, ref. [37] used IPA to explore counselling psychologists’ subjective responses to working with trauma and detailed the moral and ethical dilemmas practitioners encounter as well as the challenging interpersonal dynamics that can arise with patients who have complex histories of exposure to traumatic events. These authors also highlighted the importance of supervision in facilitating coping and the positive transformations that can occur in the clinician’s sense of self through trauma work. Another study [38] phenomenologically explored therapists’ vicarious exposure to complex traumatic material and reported on themes of self-doubt, distress, and guilt. These experiences were ascribed to the incongruity between the intervention model used by the practitioners and the needs of survivors of multiple traumatic events. The current study remains distinctive owing to its focus on the subjective experiences of female psychologists who work predominantly with sexual trauma.

The findings of this study have important implications for clinical practice. Information about the phenomenology of vicarious sexual trauma and its implications for the clinician’s sense of self and ability to respond therapeutically needs to be incorporated into the curricula of graduate training programs. This may be beneficial in preparing clinicians for trauma work by sensitizing them to the effects of working with survivors. In addition, it is imperative that supervisors and administrators facilitate the mental health of practitioners by offering a supportive environment. This can include consistently educating clinicians on the personal impact of working with sexual trauma, diversifying their caseloads, and encouraging clinicians to engage in activities that can enhance their internal capacities to tolerate working with sexual trauma (e.g., personal counselling, meditating, and journaling).

This study has certain limitations. First, the sample was small. Although the use of a small sample is consistent with the principles of IPA, replication of the results with a broader sample may be an important focus for future research. This study also focused exclusively on female psychologists. There is currently limited qualitative research on the experiences of men who treat survivors of trauma. In a systematic review, ref. [39] concluded that one of the reasons for this is that male professionals typically treat the perpetrators of abuse while female practitioners tend to treat victims. Nevertheless, the few studies that have looked at the experiences of male mental health care providers have found differences in the experience and expression of distress among men compared to women [36]. Future studies are needed to explore gender differences in the phenomenology of vicarious exposure to sexual trauma.

## 5. Conclusions

A cursory review of published South African research over the past decade reveals only one study [12] that has explored the experience of vicarious trauma among psychologists. Given the prevalence of trauma in the country, investigating the impact of trauma work on mental health care professionals remains imperative. The current study represents an important extension of previous research and highlights distinctive themes characteristic of the phenomenology of vicarious trauma among female mental health care providers who work with survivors of sexual assault.

## Figures and Tables

**Table 1 ijerph-19-03925-t001:** Brief description of themes.

Themes	Description
Shattered assumptions	Personal vulnerability, mistrust of men, lack of safety
2. Self-blame	Appraisals of complicity in doing harm, survivor guilt
3. Intrusive re-experiencing	Triggers, flashbacks, and nightmares
4. Disengagement	Sexual intimacy avoidance, emotional distancing, work-related avoidance
5. Coping resources	Modulated exposure to trauma work, instrumental support, positive re-appraisal

## Data Availability

The data presented in this study are available upon request to the second author.

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
