# Peer review of "“The Ugliness of It Seeps into Me”: Experiences of Vicarious Trauma among Female Psychologists Treating Survivors of Sexual Assault"

_ijerph, 2022, doi:10.3390/ijerph19073925_

Round 1
Reviewer 1 Report
Thank you for the opportunity to review your paper. The aim of the paper was to qualitatively explore the lived experiences of female psychologists who work with victims of sexual assault. This paper was very well written, clear, and concise. The topic is very important and your findings could have clear implications – well done!
Here are a few thoughts:
Introduction:
- It has been associated with a range of adverse mental and physical health outcomes for survivors, including post-traumatic stress disorder (PTSD), depression, substance use, anxiety, suicidality, and negative impacts on reproductive health.
- A nationally representative survey of school children [5] found that
- This same error was made on pg 1, line 42.
- Pg 2, ln 58 should read “patients’”
- Pg 2, ln 63 needs worded added, such as “the authors describe…”
- Pg 3, ln 99 repeats what has already been said – consider removing sentence.
- Pg 3, ln 101 looks like it is missing some words at the end.
- Further clarification – beyond just lack of data – about how this study is different than the previous work in this area would be helpful to explain how this study will fill in the gaps in knowledge. I understand that the research is scarce but did you do anything different in your methodology as well?
Methods:
- I would encourage you to replace Caucasian with White (eg., https://www.nature.com/articles/d41586-021-02288-x)
- You may want to consider using the COREQ as a guide to best practices when reporting qualitative data, see: http://cdn.elsevier.com/promis_misc/ISSM_COREQ_Checklist.pdf
- This would also help for the reader to understand the relationship of the researchers to the participants and situate the researchers in the data.
- Did only one person code the data?
Results:
- At times you over-rely on the quotes to explain your results. I would encourage you to provide more narrative and then use quotes to illustrate what you have found. A good article outlining this can be found here: https://journals.sagepub.com/doi/10.1177/1609406920969268?icid=int.sj-abstract.similar-articles.1
Discussion:
- In your abstract, you say “These 19 findings have important implications for clinical practice” and yet you don’t really offer too many practical implications for psychologists to consider. I would strongly suggest that you use each of the themes to identify ways in which psychologists could use this information to guide their own practice.
Author Response
We are grateful to the reviewer for the constructive feedback. Please find our responses highlighted below.
Reviewer 1
Introduction:
- It has been associated with a range of adverse mental and physical health outcomes for survivors, including post-traumatic stress disorder (PTSD), depression, substance use, anxiety, suicidality, and negative impacts on reproductive health.
Response: The sentence has been reworked.
- A nationally representative survey of school children [5] found that
- This same error was made on pg 1, line 42.
Response: This has now been corrected and reads as “A nationally representative survey of school children [5] found that…”
- Pg 2, ln 58 should read “patients’”
Response: This has been changed to: “with survivors of sexual assault…”
- Pg 2, ln 63 needs worded added, such as “the authors describe…”
Response: This has been changed to “the authors describe”
- Pg 3, ln 99 repeats what has already been said – consider removing sentence.
Response: Agreed, the two sentences have been removed.
- Pg 3, ln 101 looks like it is missing some words at the end.
Response: This sentence was deleted as it was linked to the previous sentence and was repetitious.
- Further clarification – beyond just lack of data – about how this study is different than the previous work in this area would be helpful to explain how this study will fill in the gaps in knowledge. I understand that the research is scarce but did you do anything different in your methodology as well?
Response: This last paragraph of the Introduction has been expanded to include the distinctiveness of the methodology.
Methods:
- I would encourage you to replace Caucasian with White (eg., https://www.nature.com/articles/d41586-021-02288-x)
Response: Agreed. Changed to White.
- You may want to consider using the COREQ as a guide to best practices when reporting qualitative data, see: http://cdn.elsevier.com/promis_misc/ISSM_COREQ_Checklist.pdf
- This would also help for the reader to understand the relationship of the researchers to the participants and situate the researchers in the data.
Response: This is very helpful. Thank you. We shall incorporate this checklist for future work using qualitative research. Currently, we have provided further clarity on the role of each researcher.
- Did only one person code the data?
Response: The roles of the first and second author have been further expanded upon in the Data Analysis section.
Results
- At times you over-rely on the quotes to explain your results. I would encourage you to provide more narrative and then use quotes to illustrate what you have found. A good article outlining this can be found here: https://journals.sagepub.com/doi/10.1177/1609406920969268?icid=int.sj-abstract.similar-articles.1
Response: Yes, the article does make extensive use of quotes. The main reason for this is the methodological approach which necessitates that sufficient evidence from quotes be used as evidence for the themes that have been identified. The article suggested is useful in providing a more nuanced understanding of the necessity of using quotes.
Discussion:
- In your abstract, you say “These 19 findings have important implications for clinical practice” and yet you don’t really offer too many practical implications for psychologists to consider. I would strongly suggest that you use each of the themes to identify ways in which psychologists could use this information to guide their own practice.
Response: A paragraph has been added to the Discussion section regarding the implications for clinical practice.
Reviewer 2 Report
The article “The Ugliness of It Seeps Into Me: Experiences of Vicarious Trauma Among Female Psychologists treating Survivors of Sexual Assault” written by Padmanabhanunni and Gqomfa investigates the lived experiences of those female psychologists who have treated women who were sexually abused. It is valuable the attempt to give scientific literature relevant results about this manner. Furthermore, the whole study is well orginised: the introduction section delves well into the topic, and the results are systematically structured so that the reader could easily read them.
I can easily see that you have stated a clear research question, and your initial purpose has been well deepened and concluded. Considering this study as a trustworthy and reliable one, there are few points that in my opinion should be reconsidered and much more deepened. Most importantly, the materials section as well as the discussion one, should be broadened in order to give a more detailed description of the methods which were used and the interpretations about the results.
Introduction
- This section reflects a good description of the current state of the art, reporting good incidence and prevalence rates of sexual abuse and especially the intervening psychological consequences. In addition, there are good correlations between what can happen to patient-survivors and consequently to the clinicians who are treating them. However, it would be interesting to provide the readers with additional literature about what normally happens under the same circumstances to male clinicians.
- Furthermore, you might consider implementing the reported literature with previous studies that have already used IPA methodology, looking over whether this tool has already been used to interpret similar topic.
Methods
- With regard to the methods, a good sampling methodology has been used and also, the sample size is considerable with a consequent excellent potential to get a large scale and quantity of responses, leading to valid and reliable results. Furthermore, as the good standard guidelines suggest, clinicians’ personal data have been confidentially well treated. Although, this section may be implemented.
- The description of the study design seems to be somewhat brief compared to the existing literature on the use of IPA. You might consider giving the reader a more notional description of this tool, even if it is already largely used and known.
- Since the choice of the study design is reliable and coherent with the topic, as is the analysis tool you have chosen, it would be useful to deepen the rationale behind this choice. With regard to IPA, what can it add to the state of the art on the topic of vicarious trauma? Can IPA have a practical impact on the understanding of the psychological consequences for clinicians treating patients who survived to sexual abuse?
- Regarding the participants, you might implement with a “control” group. In this particular field of qualitative research, with “control” I do not intent the technique that allows the creation of a manufactured condition, as it would cause a loss of personal information, instead I mean the creation of a subgroup, in order to make more sense and organization within the information gathered. For example, instead of having two participants alone who have experienced a prior sexual victimization, it might be more systematic to divide the group in two subgroups, one composed by those female psychologists who have an abuse precedent and the other with those who have not. The same technique, as previously underlined, it could be used to investigate the same variables within the male population as you stated at the end of the discussion section.
Discussion and limitations
- You discussion section is well treated, particularly in light of the general awareness of the limitations. The discussions are in line and coherent with the findings, which are in turn good and reliable for future research.
- Nonetheless, the only point I would implement is the bibliography. Given the relevance of your topic and the good correlation you could find with cognitive theories, the literature should be implemented with the newest references about stress and coping models. There are some studies that you have mentioned (see the cited studies published around the 90s or more than ten years ago) which date back to publications that are somewhat old and obsolete, particularly with respect to vicarious trauma. This part should be revised towards an alignment with the rest of the literature you have cited, which instead is recent and up-to-date.
Finally, I am convinced that this study is in line with the trustworthiness requested by qualitative research, so that you can eventually fix it with minor revisions.
Author Response
We are grateful to the reviewer for the constructive feedback. Please find our responses highlighted below.
Reviewer 2
Introduction
- This section reflects a good description of the current state of the art, reporting good incidence and prevalence rates of sexual abuse and especially the intervening psychological consequences. In addition, there are good correlations between what can happen to patient-survivors and consequently to the clinicians who are treating them. However, it would be interesting to provide the readers with additional literature about what normally happens under the same circumstances to male clinicians.
Response: Agreed, this is a relevant point. In reviewing the literature, we identified a dearth of studies focusing on gender differences in susceptibility to vicarious trauma. This was confirmed in a systematic review on gender differences by Baum (2016) who concluded that there is an insufficient number of studies that have examined the responses of male mental health care providers to vicarious exposure to trauma but that the few existing studies suggest that men display vicarious trauma responses differently to women. This information has now been incorporated into the limitations section of the study rather than the introduction.
- Furthermore, you might consider implementing the reported literature with previous studies that have already used IPA methodology, looking over whether this tool has already been used to interpret similar topic.
Response: Studies using a similar methodological approach have been included in the introduction (e.g., 12). The Discussion section has now been expanded and highlights the contribution of the existing study to prior work using IPA.
Methods
- The description of the study design seems to be somewhat brief compared to the existing literature on the use of IPA. You might consider giving the reader a more notional description of this tool, even if it is already largely used and known.
Response: The section on IPA in the introduction has been expanded upon to highlight the contribution of the methodology towards expanding the literature base. In addition, the process of data analysis using IPA has now been expanded upon.
- Since the choice of the study design is reliable and coherent with the topic, as is the analysis tool you have chosen, it would be useful to deepen the rationale behind this choice. With regard to IPA, what can it add to the state of the art on the topic of vicarious trauma? Can IPA have a practical impact on the understanding of the psychological consequences for clinicians treating patients who survived to sexual abuse?
Response: Please see previous response.
- Regarding the participants, you might implement with a “control” group. In this particular field of qualitative research, with “control” I do not intent the technique that allows the creation of a manufactured condition, as it would cause a loss of personal information, instead I mean the creation of a subgroup, in order to make more sense and organization within the information gathered. For example, instead of having two participants alone who have experienced a prior sexual victimization, it might be more systematic to divide the group in two subgroups, one composed by those female psychologists who have an abuse precedent and the other with those who have not. The same technique, as previously underlined, it could be used to investigate the same variables within the male population as you stated at the end of the discussion section.
Response: This would provide for an interesting comparison and will be considered for a future study.
Discussion and limitations
- You discussion section is well treated, particularly in light of the general awareness of the limitations. The discussions are in line and coherent with the findings, which are in turn good and reliable for future research.
Response: Thank you for the positive feedback.
- Nonetheless, the only point I would implement is the bibliography. Given the relevance of your topic and the good correlation you could find with cognitive theories, the literature should be implemented with the newest references about stress and coping models. There are some studies that you have mentioned (see the cited studies published around the 90s or more than ten years ago) which date back to publications that are somewhat old and obsolete, particularly with respect to vicarious trauma. This part should be revised towards an alignment with the rest of the literature you have cited, which instead is recent and up-to-date.
Response: The Discussion section has been expanded and includes more current literature on the topic. The references from the 1990’s are seminal works. For example, McCann & Pearlman’s (1990) paper on vicarious trauma represents a seminal study. Similarly, the study by Schauben, L. J., & Frazier, P. (1995) is one of the first studies on vicarious trauma among female counsellors working with sexual assault victims and is still regarded as a seminal paper.
Round 2
Reviewer 1 Report
Thank you for your efforts to address the reviewer comments - I think you have adequately responded to the feedback.